# Changes in Cellular Regulatory Factors before and after Decompression of Odontogenic Keratocysts

**DOI:** 10.3390/jcm10010030

**Published:** 2020-12-24

**Authors:** Slmaro Park, Han-Sung Jung, Young-Soo Jung, Woong Nam, Jung Yul Cha, Hwi-Dong Jung

**Affiliations:** 1Department of Oral & Maxillofacial Surgery, College of Dentistry, Yonsei University, 50-1 Yonsei-Ro, Seodeamun-Gu, Seoul 03722, Korea; kingpsmr@naver.com (S.P.); ysjoms@yuhs.ac (Y.-S.J.); omsnam@yuhs.ac (W.N.); 2Division in Anatomy and Developmental Biology, Department of Oral Biology, Oral Science Research Center, BK21 PLUS Project, Yonsei University College of Dentistry, Seoul 03722, Korea; hsjung@yuhs.ac; 3Department of Orthodontics, College of Dentistry, Yonsei University, 50-1 Yonsei-Ro, Seodeamun-Gu, Seoul 03722, Korea

**Keywords:** IHC staining, immunohistochemical staining, Bcl-2, EGFR, Ki-67, PCNA, P53, SMO, decompression, enucleation, odontogenic keratocysts

## Abstract

Decompression followed by enucleation, which is one of the treatments used for odontogenic keratocysts (OKCs), is frequently used in OKC lesions of large sizes. This method offers the advantage of minimizing the possibility of sensory impairment without creating a wide-range bone defect; moreover, the recurrence rate can be significantly lower than following simple enucleation. This study aimed to assess the changes in histology and expression of proliferation markers in OKCs before and after decompression treatment. A total of 38 OKC tissue samples from 19 patients who had undergone decompression therapy were examined morphologically and immunohistochemically to observe changes in proliferative activity before and after decompression. The markers used for immunohistochemistry (IHC) staining were Bcl-2, epidermal growth factor receptor (EGFR), Ki-67, P53, PCNA, and SMO. The immunohistochemistry positivity of the 6 markers was scored by using software ImageJ, version 1.49, by quantifying the intensity and internal density of IHC-stained epithelium. The values of Bcl-2, Ki-67, P53, proliferating cell nuclear antigen (PCNA), and SMO in OKCs before and after decompression showed no significant change. No correlation between clinical shrinkage and morphologic changes or expression of proliferation and growth markers could be found. There was no statistical evidence that decompression treatment reduces potentially aggressive behavior of OKC within the epithelial cyst lining itself. This might indicate that decompression does not change the biological behavior of the epithelial cyst lining or the recurrence rate.

## 1. Introduction

The odontogenic keratocyst (OKC) is a distinctive form of developmental odontogenic cyst with potential for aggressive and infiltrative behavior that originates from the dental lamina remnants or from the basal cells of overlying epithelium [1,2]. OKCs have a predilection for the posterior part of the mandible, a peak incidence in patients between second and fourth decades, and a slight male predominant tendency. Radiographically, the OKC is shown as a unilocular or multilocular lesion, most often surrounded by smooth or scalloped margins with sclerotic borders [1] (Figure 1).

The OKC shows aggressive clinical behavior with a high and extremely varied recurrence rate [3,4,5]. The recurrence rate is reported to be 25–60%, varying according to the treatment method [1]. The OKC typically has a thin, friable wall, which is often difficult to enucleate from the bone in one piece. The epithelial lining is composed of a uniform layer of stratified squamous epithelium, usually 6 to 8 cells in thickness. The luminal surface shows flattened parakeratotic epithelial cells, which exhibit a corrugated appearance. The basal epithelial layer is composed of a palisaded layer of cuboidal or columnar epithelial cells, which are often hyperchromatic [1].

In 2005, the World Health Organization (WHO) renamed it “keratocystic odontogenic tumor” (KCOT), reclassifying it among the epithelial odontogenic tumors [6] due to studies which supported the hypothesis that OKCs are a neoplastic condition [7,8]. This assumption was based on the fact that the lesion shows overexpression of proliferative markers and Bcl-2, in addition to loss of heterozygosity or methylation of tumor suppressor genes and mutation of the *PTCH1* gene [7]. However, in 2017, the WHO moved the keratocystic odontogenic tumor classification from the neoplastic category (2005) back into the cyst category (2017), under the category of developmental odontogenic cysts. The 2017 classification reverted to the original and well-accepted terminology of the OKC because many papers showed that the *PTCH* gene mutation could be found in non-neoplastic lesions, including dentigerous cysts; furthermore, many researchers suggested that resolution of the cyst after marsupialization was not compatible with a neoplastic process. It is, however, still important and clinically relevant to separate the OKC from the other odontogenic cysts due to its diagnostic histologic features as well as its distinctive clinical features [9].

Its aggressive clinical behavior and frequent recurrence following curettage has been the focus of several studies which indicate that the OKC epithelial lining may have some intrinsic growth potential [8,10,11,12,13,14]. The proliferative potential can be assessed by immunohistochemistry using monoclonal antibodies against specific cell cycle associated proteins [15,16,17].

Various treatment modalities are reported which can be generally classified as conservative or aggressive. Conservative treatment usually includes enucleation and/or marsupialization, while aggressive treatment includes enucleation associated with adjunct therapies or resection [18,19,20]. The reported frequency of recurrence in various studies ranges from 5% to 62%. This wide variation may be related to the total number of cases studied, the length of follow-up periods, and the inclusion or exclusion of orthokeratinized cysts in the study group [1].

The recurrence rates of rather conventional therapies such as enucleation and curettage are reported to be highest [21,22]. In order to reduce the recurrence rates, mass excision including adjacent normal bone or use of Carnoy’s solution and cryotherapy should be considered, but such methods can lead to complications such as bone loss and sensory impairment [23,24,25]. It is known that in the case of OKCs, mass excision or enucleation after decompression can minimize the possibility of sensory impairment without creating a wide-range bone defect; moreover, the recurrence rate can be significantly lower than that of simple enucleation [17,26]. However, given recent published reports of rather contrary results, the recurrence rate is controversial.

The development of new surgical methods and drugs to reduce the recurrence rate is urgent, but systemic research is difficult due to the low rate of morbidity. These treatments are still being reported only at the level of the presentation in case reports [27], with little research on changes in growth factors or cell apoptotic factors after decompression.

According to previous studies, proliferation factors of OKCs before and after decompression have shown inconsistent and different results. Thus, the present study hypothesized that expression of the proliferation factors would decrease. The surgeons of the Department of Oral and Maxillofacial Surgery had previously performed decompression and incisional biopsy on OKC lesions and later obtained tissue specimens when performing enucleation. Using these specimens, the present study aimed to investigate and identify, through histology and expression, proliferation markers in OKCs before and after decompression treatment through immunohistochemical staining methods.

## 2. Materials and Methods

### 2.1. Materials

A total of 38 formalin-fixed paraffin-embedded tissue samples of OKCs were retrieved from the files of the Department of Oral Pathology, College of Dentistry, Yonsei University. The samples met the inclusion criteria of histologically-confirmed OKC which had been subjected to decompression treatment, followed by enucleation on the same lesion at later on. Cases of OKCs associated with basal cell nevus syndrome and orthokeratinized OKCs were excluded. Of the total, 9 patients had lesions on their maxillae, and 10 patients had lesions on their mandibles, as summarized in Table 1. All patients underwent the treatment at the Department of Oral and Maxillofacial Surgery, College of Dentistry, Yonsei University from 2013 to 2018. A polyethylene tube was inserted and fixed to the lesions at the time of primary incision biopsy. Patients were instructed to irrigate the cystic cavity with normal saline or clean water on a daily basis. The patients underwent periodic clinical and radiographic examination of the lesion to observe progression at an average interval of several months. Specimens for histology and pathology were taken during the decompression procedure for confirmation of diagnosis and subsequent enucleation. These surgical specimens constituted the materials upon which this investigation is based. The present research has passed deliberation of the Institutional Review Board (approval number 2-2018-0050.)

### 2.2. Immunohistochemistry (IHC) Staining

#### 2.2.1. Markers

Bcl-2

Bcl-2 is located in the external membrane of mitochondria as well as the endoplasmic reticulum and nuclear membrane. Overexpression of Bcl-2 in tumor cells causes apoptotic resistance and increased cell growth [28]. Uncontrolled expression of Bcl-2 has been found before histopathologic changes in the early stage of neoplasm [29,30].

B.EGFR

Epidermal growth factor receptor (EGFR) is the most important growth factor ligand on the cell surface. EGFR is involved in various cellular processes, including cell growth, motility, inhibition of apoptosis, and cell adhesion [31,32].

C.Ki-67

Ki-67 protein is a 319–358 kDa alternatively-spliced amphophilic non-histone nuclear protein that consists of multiple unique Ki-67 motifs. Ki-67 is an absolute requirement for DNA synthesis [33].

D.P53

P53 is produced by the tumor suppressor gene P53 effective at the G1 phase of the cell cycle, and participates in growth arrest, initiates repair, or induces apoptosis [34]. Detection of this marker immunologically may indicate stabilization of this protein and reflects cell cycle regulation in favor of proliferation [35].

E.PCNA

Proliferating cell nuclear antigen (PCNA) is a 36 kDa acidic non-histone nuclear protein important for DNA synthesis and repair. In the presence of replication factor C, a multi-subunit complex, PCNA allows DNA polymerase to initiate leading-strand DNA synthesis [36]. Immunostaining with a monoclonal antibody (PC10) against this antigen has been shown to demonstrate the proliferative compartment of normal tissue [37] and to correlate well with prognosis in some tumors.

F.SMO

The hedgehog (HH) signaling pathway is a key regulator of embryonic development, controlling both cellular proliferation and cell fate. Binding of sonic hedgehog (SHH) to its receptor, patched (PTCH1), is believed to relieve normal inhibition by PTCH1 of smoothened (SMO), a seven-span transmembrane protein with homology to a G-protein-coupled receptor r [38]. The protein generated by SMO is downstream of PTCH1; that is, the expression of PTCH1 restrains the activation of SMO, and thereby inhibits activation of the HH pathway [39,40].

#### 2.2.2. Staining Method

In order to identify the expression of cancer markers in tissues obtained at decompression and enucleation, immunohistochemistry staining using antibodies of Bcl-2 mouse mAb (Cell Signaling Technology Inc., Danvers, MA, USA, #15071), EGFR rabbit mAb (Cell Signaling Technology Inc., #4267), P53 mouse mAb (Cell Signaling Technology Inc., #48818), Ki-67 rabbit mAb (Cell Signaling Technology Inc., #9027), PCNA rabbit mAb (Cell Signaling Technology Inc., #13110), and SMO mouse mAb (Santa Cruz Biotechnology Inc., Dallas, TX, USA, sc-166685). Formalin fixed tissue was made into paraffin block after the flush, dehydration, transparency, and paraffin penetration process. The paraffin fragments were made and dried by stripping the tissue to a constant thickness (4 µm). To conduct the immunohistochemical examination, the paraffin of each tissue section was removed using xylene and then hydrated, hydrogen peroxide (3% H_2_O_2_) then being added to inhibit intrinsic enzymes for 10 min at room temperature. In order to restore the cross-linking induced by formalin fixation to its original state, slices were put into the citrate buffer and boiled for 10 min in the microwave to perform heat antigen retrieval. The boiling slices were cooled at room temperature while immersed in a citrate buffer and treated at 5% BSA for an hour to suppress non-specific proteins. The diluted primary antibodies were then treated with incision and incubated at 4 °C for overnight. The dilution concentration of Bcl-2 and Ki-67 antibodies was 1/500, P53, EGFR, SMO antibodies 1/250, and PCNA antibodies 1/5000, achieved by diluting them in a 5% BSA solution. After primary antibody processing and washing, the secondary antibody with biotin and polymer-HRP were applied at room temperature for 10 min following the protocol using the Polink-2HRP plus road 3, 3′-Diaminobenzidine (DAB) detection system (GBI Labs Inc., Bothell, WA, USA, D41-18) and the color was manifested. Hematoxylin was used for contrast dyeing. After the primary antibody treatment, the phosphate-buffered solution with Tween was used to flush three times at room temperature for five minutes between each process. Dyeing-completed tissue slices were dehydrated, sealed into permounts, and observed under a microscope. The analysis was done by using the ImageJ program to measure the histogram and densities of the dyed parts of DAB.

### 2.3. Staining Counting Method

#### 2.3.1. Measuring Intensity

Using the ImageJ program, the original image of RGB (red, green, blue) was adjusted to 8-bit color, and the threshold then set. The histogram was generated in 3 fields to measure the intensity within a certain area (Figure 2). This method was used to quantify the staining of several photos per sample.

#### 2.3.2. Measuring Internal Density

Using the ImageJ program, the external part of the epithelium was first removed from the original stained photo, after which the color threshold was established. Statistics were analyzed measuring the internal density within a certain area in 3 fields (Figure 3). In addition to the method using intensity, internal density was used to quantify the staining in several stained photos per sample.

### 2.4. Statistical Analysis

Statistical analyses were performed to compare the staining values obtained from decompression specimens with those of enucleation tissues. Quantified values using intensity showed a Gaussian distribution. The paired t-test was used to compare value changes among the groups (SPSS^®^ 25, IBM, Armonk, NY, USA). Most quantified values using internal density were considered to lack a Gaussian distribution. The Wilcoxon signed ranks test was used to compare value changes among the groups (SPSS^®^ 25, IBM, USA). Statistical difference between the maxilla group and the mandible group was also determined. *P*-values less than 0.05 were considered significant differences.

## 3. Results

### 3.1. Decompression and Follow Up

A total 19 patients, 9 female and 10 male, underwent the decompression procedure. The patients ranged between 19 to 81 years old, the average age being 38.8. Of them, 9 patients had OKCs on their maxillae. A total of 7 patients had 3rd molars with the lesion and 3 of them had a 3rd molar inside the sinus due to OKC enlargement. Third molars were removed during enucleation surgery. A total of 10 patients had OKCs on their mandibles. A total of 3 patients had OKC lesions on their rami, 1 of them with the lesion involved in condyle head to ramus. In addition, 2 patients had 3rd molars within the OKC lesion, which were also removed when enucleation was performed. The progress after decompression was certain. The size of the lesions had reduced, and the bone density on panorama x-ray was observed to have thickened. The decompression periods of the patients ranged between 4 and 12 months, the average period being 7.3 months. After decompression, enucleation was performed. OKCs recurred among 3 patients, all of whom had OKCs on maxilla. Of the total, 2 patients experienced OKC recurrence 24 months after enucleation, and OKC recurred in 1 patient 10 months after enucleation. Patients were all instructed to visit the Department of Oral and Maxillofacial Surgery for 10 years due to the characteristic high recurrence rate. However, 6 patients were lost in follow-up.

### 3.2. Immunohistochemical Analysis

The OKC tissues showed differences before and after decompression. The cystic cavity wall was constructed of fibrous tissue covered by very thin regular parakeratinized stratified squamous epithelium consisting of 5 to 8 layers. The basal cells consisted of columnar cells and palisading nuclei. Many epithelia were separated from the fibrous capsule. There was very loose connection between the epithelium and connective tissue without rete pegs. The tissues obtained at the time of enucleation had changed. The epithelium of the cyst wall had become hyperplastic stratified squamous epithelium and dense connective tissue with infiltration of inflammation cells. Bcl-2 staining was expressed in the basal layer of the epithelium. However, in some samples, Bcl-2 positive cells were observed in all layers of the lining epithelium and connective tissue cells of OKCs (Figure 4). EGFR showed membranous and cytoplasmic staining of epithelial cells, progressively diminishing from the basal toward the superficial layers. Its expression was prominent in basal epithelial cells (Figure 5). Ki-67 positive cells were mainly distributed in the basal and parabasal layers of the epithelium. Photos showed a tendency toward less staining of Ki-67 in enucleation tissues. However, quantified values did not show the same tendency before and after decompression (Figure 6). P53 showed abundant expression in both basal and suprabasal areas (Figure 7). PCNA positive cells were seen throughout the epithelium but mainly in the parabasal layers (Figure 8). Positive immunostaining of SMO was detected in the intermediate layer, but rarely in the superficial and basal layer (Figure 9). Bcl-2, Ki-67, P53, and PCNA staining were confined to the epithelial cell nucleus of the OKC. EGFR and SMO staining was confined to the cell membrane of the epithelium. The positive cells of 6 markers after decompression showed similar patterns.

### 3.3. Statistical Results

The quantified values of six markers showed both increase and decrease without displaying a tendency before and after decompression. Statistic evaluation was performed to find significant differences in the values. The paired *t*-test was applied on numerical parameters of 6 markers obtained by quantifying the IHC staining intensity. The EGFR values before and after decompression yielded a *p*-value of 0.040, considered significant. However, values of P53, SMO, Bcl-2, Ki67, and PCNA before and after decompression resulted in *p*-values of 0.370, 0.373, 0.785, 0.678, and 0.271 respectively, which were considered insignificant. Statistics of EGFR values showed no significant difference between maxilla and mandible.

Internal density was also used to quantify IHC staining. Numerical values obtained for the 5 markers P53, SMO, Bcl-2, Ki-67, and EGFR did not meet the test of normality. Statistics performed using the Wilcoxon signed ranks test yielded *p*-values of 0.968, 0.372, 0.147, 0.355, and 0.904 respectively. The paired t-test was used on numerical parameters of PCNA, yielding a *p*-value of 0.781. No *p*-values were considered significant.

## 4. Discussion

The object of this study was to assess the changes in histology and expression of proliferation markers in OKCs before and after decompression treatment. Immunohistochemical analysis studies of OKCs have demonstrated their aggressive character. Furthermore, many studies have suggested that the epithelium adjacent to surrounding bone may be related to the aggressive character and proliferative activity of OKCs [8,10,11,12,13,14]. If the assumption is correct, rupture in the continuity of the epithelium when performing decompression may decrease cyst size and mitigate proliferation. Interestingly, a study in which OKC walls were transplanted into athymic mice demonstrated that the features of the epithelial lining were only maintained in the presence of its cystic wall [41]. There are markers relative to proliferation in the epithelium of OKCs; if rupture of the epithelium reduces proliferation of OKCs, observation of associated markers before and after decompression may shed light on the relation.

Although intensity and internal density were utilized to quantify the positivity of IHC staining, the results showed no statistical significance. The values of EGFR before and after decompression were found to be significantly different using the intensity method. However, this method consistently reflected the background colors to the quantified values. Hence, the internal density method was assumed to be more precise. The values of the 6 markers before and after decompression obtained using the internal density method showed no significant differences. Other studies have calculated the number of positive cells per unit length of epithelial basement membrane (BM) or the fraction of cycling cells within the investigated cell population at a given time (known as proliferation of labeling index). Studies using these methods usually involve two or more pathologists. Since it was not possible to apply those methods, expression of proliferation markers was analyzed with the aid of ImageJ software version 1.49.

Here is the specific information about the markers used in this study. The Bcl*-2* gene has the ability to inhibit apoptosis without encouraging cell proliferation leading to cell cycle changes which facilitate cell survival independent of cell division. The expression of Bcl*-2* gene is connected with low-grade tumors and its inhibition of apoptosis is regarded as a common tumor genesis pathway [42]. Lin et al., have demonstrated that the localization of EGFR is correlated with a highly proliferative status of tissues, and have provided evidence to support the potential roles of EGFR as a transcription factor or coactivator, which might activate the genes required for its mitogenic effects. The location of this receptor within a cell may be related to its response to proliferative stimuli. Cells proliferating at a physiologic rate express this receptor in both membrane and cytoplasm [43]. The immunolocation of EGFR in odontogenic epithelium may therefore be associated with the origin of odontogenic cysts and tumors [44]. Overexpression of EGFR-related genes, seen in many neoplasms, causes the oversensitivity of cells to a normal level of growth factor. Nowadays, EGFR is known as an effective growth factor in many human cancers [28]. EGFR signaling is associated with malignancy transformation, giving rise to specific phenotypes of cells which can affect the cellular reaction to the treatment [44]. Ki-67 is known as a marker of cell proliferation because the Ki-67 antigen is preferentially expressed in proliferative cells during late G1, S, G2, and M phases, whereas resting, noncycling cells (G0 phase) lack Ki-67 expression. Ki-67 thus immunohistochemically provides a reliable index of cellular proliferation [45]. Although its level increases during the S-phase, MIB-1 recognizes Ki-67 antigen in the entire cell cycle [10,46]. Because of its absence in quiescent cells (G0 phase), this protein has developed into a widely-used tumor marker in the fields of research and pathology [47]. Ki-67 is of prognostic value for many types of malignant tumors [48]. The P53 gene, located on chromosome 17q13, encodes a nuclear phosphoprotein which is thought to control cell growth at the G1/S checkpoint. SMO is a tumor-related gene located at 7q32.3. It contains 12 exons spanning approximately 24 kb and encodes a 787-amino-acid transmembrane glycoprotein [49]. Its receptor is a G protein-coupled receptor that interacts with Patched, an important part of the HH signaling pathway during embryogenesis as well as adulthood [50,51]. The HH pathway has been demonstrated to play an important role in different development-related cancers [52,53], but the exact mechanism of action has not yet been elucidated.

The markers used in the present study have been used in several other studies comparing their expression in OKCs with other lesions. The apoptosis-related factors P53 protein and Bcl-2 protein have been found in the lining epithelium of OKCs [54,55,56]. Mendes et al., observed mild to strong expression of COX-2 in all of 20 (100%) cases. Fifteen (75%) cases of OKC stained positive for P53 and 18 (90%) stained positive for Ki-67. There were no statistically relevant differences among the expressions of COX-2, Ki-67, and P53 [57]. However, Slootweg has found the presence/absence of densely-stained P53 positive cells to be broadly related to Ki-67 cell numbers in highly proliferative areas as well as the converse [58]. The present study did not find any relation between expression of P53 and Ki-67.

PCNA, Ki-67, and P53 protein were all expressed in actively proliferating cells, particularly in neoplasms. They were expressed more strongly in OKCs than in other odontogenic cysts and more particularly so in the OKCs associated with nevoid basal cell carcinoma syndrome (NBCCS) [13]. Studies comparing OKCs and dentigerous cysts have demonstrated a greater proliferative potential of the OKC epithelial lining comparable to that of ameloblastoma [10,46]. P53 was found in OKCs more often than in other odontogenic cysts [58] or only in OKCs [12], suggesting that increased epithelial activity explains the tendency to recur. In the present study, patients 7 and 9 showed increased expression of P53 after decompression, but patient 4 had decreased expression of P53.

Merva et al., compared expressions of bax, Bcl-2, and Ki-67 in OKCs, ameloblastomas, and radicular cysts. Ameloblastomas showed stronger Bcl-2 expression than OKCs and radicular cysts. Bcl-2 expression in OKCs was significantly higher than in radicular cysts. The lining epithelium of OKCs showed stronger Ki-67 expression than that in ameloblastomas and radicular cysts. It was concluded that high expression of Bcl-2 and Ki-67 in OKCs accords with its aggressive clinical behavior and high recurrence rate [59]. Kichi et al., also reported that Bcl-2 is seen only in OKCs and not in dentigerous cysts [60]. In a study done by Shear, expression of the EGFR marker was reported in the epithelium of OKCs and dentigerous as well as radicular cysts. The strongest reaction was related to OKCs, and the weakest was in radicular cysts [14]. MG de Oliveira et al., analyzed immunolabeling of Ki-67, EGFR, and Survivin in the basal and suprabasal layers of OKCs, dentigerous cysts, and pericoronal follicles. OKCs showed the highest proliferation rate among the three groups, with Ki-67 staining found mainly in suprabasal layers. EGFR immunolabeling was observed mainly in the cytoplasm in basal and suprabasal layers of OKCs [61]. Razavi et al., compared expression of Bcl-2 and EGFR in OKCs with that in dentigerous cysts and ameloblastomas. All cases of ameloblastoma and OKC, but no dentigerous cyst cases, were positively stained for Bcl-2. Expression of Bcl-2 was higher in the peripheral layer of ameloblastomas and the basal layer of OKCs. Furthermore, all cases of ameloblastoma and dentigerous cysts, but no OKC samples, were positively stained for EGFR. Expression of EGFR was higher in the peripheral layer of ameloblastomas and the basal layer of dentigerous cysts. It was concluded that the biological activity and growth mechanisms of OKCs are different compared with those of other cystic lesions [62]. In the present study, both Bcl-2 and EGFR were expressed in OKC tissues.

K. Ohki found immunoreactivity for SHH and GLI-1 was markedly higher in epithelial components than in subepithelial cells, while immunoreactivity for PTC and SMO was similar in epithelial components and subepithelial cells in OKCs. The positive rate of PTC and SMO expression in subepithelial cells of OKCs were significantly higher than those in gingiva. The positive rate of GLI-1 expression in subepithelial cells of BCNS-associated OKCs were significantly higher than those in primary OKCs. It was concluded that SHH signaling might be involved in the pathophysiologic nature of OKCs [63]. In the present study, SHH was tested for IHC staining with the tissue samples, however, staining with SHH did not work. SMO was thus used for the study instead of SHH.

T. Yagyuu et al., reported the immunoreactivity of proliferation-related SMO in OKCs with recurrence was higher than that without recurrence, whereas the expressions of a ligand, SHH, and an inhibitory receptor, Patched, were not associated with OKC recurrence. The expressions of SHH and SMO showed inverse correlation in whole OKCs. It was concluded that recurrence of the OKC is associated with multilocular large lesions and high SMO expression [64]. In the present study, however, no relationship was found between high expression and recurrence in the values obtained from the three patients who experienced recurrence.

Odontogenic tumors were studied for the markers related to the SHH signal pathway. L. Zhang et al., suggested SHH, PTC, SMO, and GLI-1 proteins are predominantly epithelial expressions of the SHH signaling pathway in odontogenic tumors. Immunoreactivity for SHH, PTC, SMO, and GLI-1 was detected in both epithelial-derived odontogenic tumors and epithelial-mesenchymal-derived odontogenic tumors with or without dental hard tissue formation. Mesenchymal-derived odontogenic tumors showed no positive staining except for the focal epithelial cells in island or cord forms within the central portion of the tumor. SHH, PTC, SMO, and GLI-1 were detected more in the cytoplasm of the epithelial cells than in stromal cells. Immunoreactivity for GLI-1 was also detected in the base membrane of the tumor cells [65].

There are controversies over some staining factors in the context of decompression treatment. Ninomiya et al., investigated Ki-67 labeling index and expression of IL-1 alpha mRNA following decompression treatment in OKCs. Their results showed a decrease in Ki-67 labeling after decompression. They concluded that proliferation activity in the lining could also be affected by changes in the intra-luminal pressure and cytokine concentrations [66]. Nakamura et al., found that the average Ki-67 labeling index after marsupialization was slightly lower than that before marsupialization, though these results were statistically insignificant [67]. S. Awni et al., found no statistical changes in Ki-67 expression before and after the decompression procedure or with inflammation [68]. The values of Ki-67 expression in the present study showed both increase and decrease after decompression, the results being statistically insignificant.

In an immunohistochemistry study, S. Awni et al., observed an increase in P53 expression with prolonged duration of treatment and in cases showing more inflammation. The same study found no statistical difference in expression of the anti-apoptotic protein Bcl-2 after treatment [68]. In the present study, inflammation cells were observed to increase in H&E-stained photos of some cases after decompression, but quantification or statistic evaluation was not performed.

Pogrel et al., reported 10 cases of OKC resolved completely with marsupialization therapy alone, Bcl-2 staining being negative in all the samples taken of resolved OKCs compared to the high expression observed in the pre-treatment biopsy [69]. Bcl-2 staining was positive in the tissues after decompression in the present study. However, this discrepancy may be due to differences in decompression and marsupialization, similarities notwithstanding.

Pia Clark et al., examined expression of P53, Ki-67, and EGFR in OKCs before and after decompression. They observed no significant change in expression values nor correlation between the expression of Ki-67 and P53. No correlation between clinical shrinkage and morphologic change or between expression of proliferation and growth markers was found. Expression of EGFR was large and had a tendency to increase after decompression. OKCs showed a high degree of EGFR expression that could indicate considerable growth potential of these. Of the 16 cysts, 13 (81.3%) showed expression before decompression, the number increasing to 15 of 16 (93.8%) after decompression [70]. Expression of EGFR in the present study showed significant difference after decompression based on the values obtained by quantifying the intensity. However, no statistical difference was found in the other values, obtained by quantifying the internal density.

Studies regarding various treatments of OKCs have been conducted. Resection was found to have the lowest recurrence rate (0%) but the highest morbidity rate. Simple enucleation was reported to have a recurrence rate of 17% to 56%. Simple enucleation combined with adjunctive therapy, such as the application of Carnoy’s solution or decompression before enucleation, was reported to have a recurrence rate of 1% to 8.7%. Resection or enucleation with adjunctive therapy was associated with recurrence rates that were lower than those associated with enucleation alone [71]. Other studies showed additional adjunctive therapies including liquid nitrogen and peripheral ostectomy. These adjuncts all described above were supposed to eliminate epithelial islands and microcysts in the peripheral bone and decrease the recurrence rates [23,72].

Some studies have suggested that pathological changes after decompression are caused by the introduction of inflammation into the OKC via decompression opening [54,73]. In addition, some have suggested decompression therapy reduces the intra-lesional pressure. Other studies have found certain changes after treatment with decompression, including thickening of the cystic wall [26], inhibition of IL-1α [66], epithelial dedifferentiation and loss of cytokeratin-10 production [74], changes for Forsell and Sainio group Ia (parakeratotic type) to group II and III (orthokeratotic type) [17], and Bcl-2 negativity [69]. It has been suggested that those changes might account for less aggressive behavior with decreased cyst recurrence after decompression [17,26].

It seemed that the same treatment yielded different immunohistochemical expressions of Bcl-2, EGFR, Ki-67, P53, PCNA, and SMO. Although IHC staining is possible in OKCs, quantification of staining results seems uncertain. Clinically, shrinkage of the cysts after decompression could be observed, no matter what the histologic picture showed.

There were studies that supported the reliability of numericalization of IHC using software such as ImageJ. Fuhrich D.G. et al., compared intraobserver and interobserver variation between traditional histological score (HSCORE) and digital HSCORE (D-HSCORE) performed by expert and naive researchers. Immunohistochemical analysis of β3 integrin subunit of 100 endometrial biopsies obtained from the midluteal phase of the menstrual cycle were reanalyzed using ImageJ software (D-HSCORE). The study concluded the D-HSCORE performed by an inexperienced researcher has high correlation to traditional HSCORE performed by an expert [75]. Similar results were yielded in other studies that compared manual counting with quantification using computer programs [76,77,78].

The results of the present study may be due to small sample size, tissues being harvested 4 to 5 years ago, and uncontrolled variation in staining conditions, as well as the smaller tissue sizes harvested in decompression relative to mass excisions. There was considerable difference in the proliferation rate even in epithelium of the same appearance. Quantifying IHC staining from another area of the cyst may have yielded a totally different result. The decompression time was determined by the clinical response, which varied and was relatively short in some cases.

## 5. Conclusions

The values of Bcl-2, Ki-67, P53, PCNA, and SMO in OKCs before and after decompression showed no significant change. Expression of EGFR values changed significantly after decompression. However, the results are uncertain because statistic confirmation was only done with the values obtained from quantification based on intensity. The values of EGFR obtained based on internal density showed no significant change. No correlation between clinical shrinkage and morphologic changes nor between expression of proliferation and growth markers could be found. According to the above results, the original hypothesis was rejected. There was no statistical evidence that decompression reduces the aggressive behavior of OKCs within the epithelial cyst lining itself. This might indicate that decompression does not change the biological behavior of the epithelial cyst lining or the recurrence rates.

## Figures and Tables

**Figure 1 jcm-10-00030-f001:**
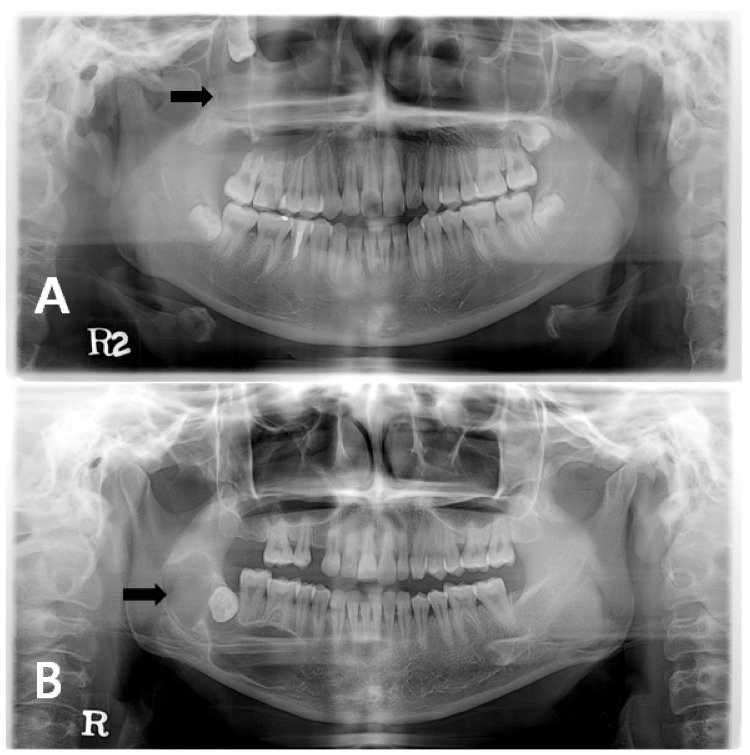
Radiographic image of odontogenic keratocyst (OKC): (**A**) patient with an OKC on maxilla, right. Third molar is floating inside sinus due to OKC; (**B**) patient with an OKC on mandible, right. It is showing scalloped margin with sclerotic border.

**Figure 2 jcm-10-00030-f002:**
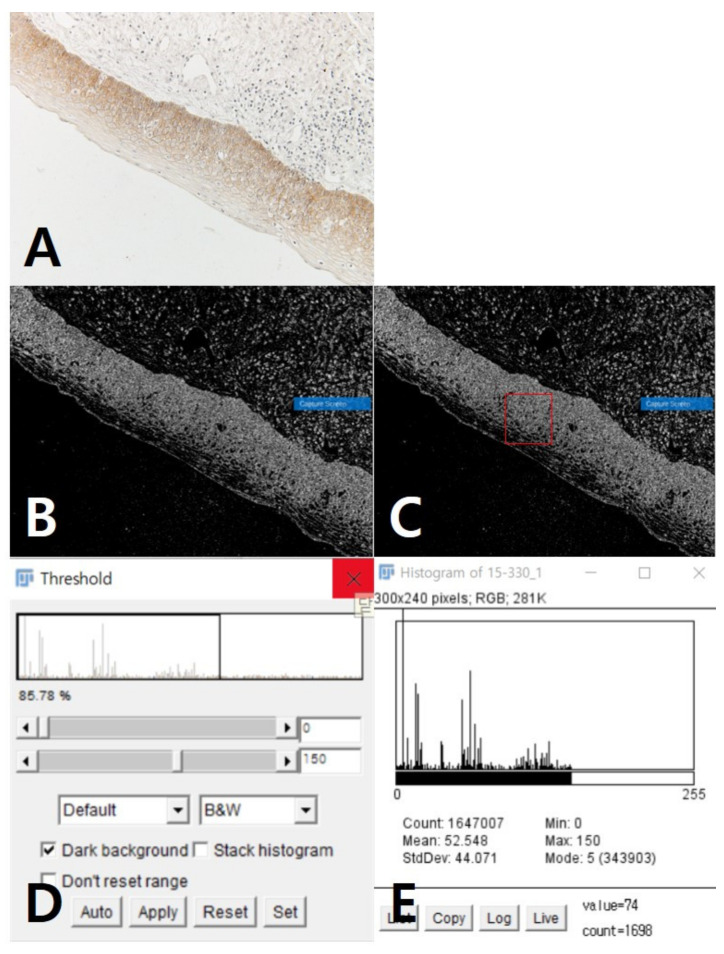
Measurement of immunohistochemistry (IHC) staining based on intensity: (**A**) original stained photo with red, green, and blue (RGB) adjusted to 8-bit color; (**B**) setting threshold; (**C**) measuring intensity within area; (**D**) threshold control window; (**E**) measured intensity in histogram form.

**Figure 3 jcm-10-00030-f003:**
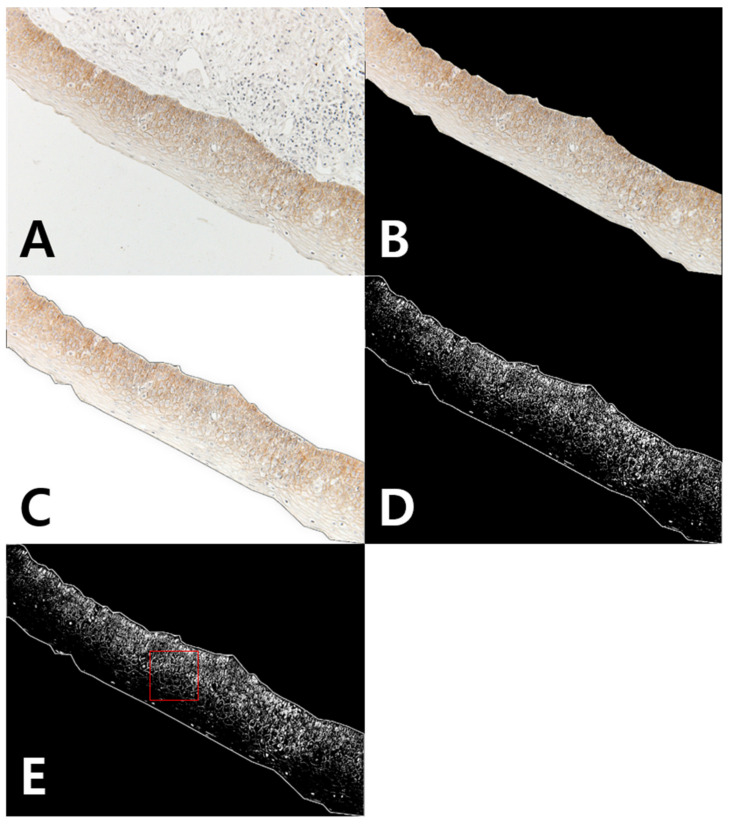
Measurement of IHC staining based on internal density: (**A**) original stained photo; (**B**) clearing outside; (**C**) subtracting background; (**D**) establishing color threshold; (**E**) measuring internal density within area.

**Figure 4 jcm-10-00030-f004:**
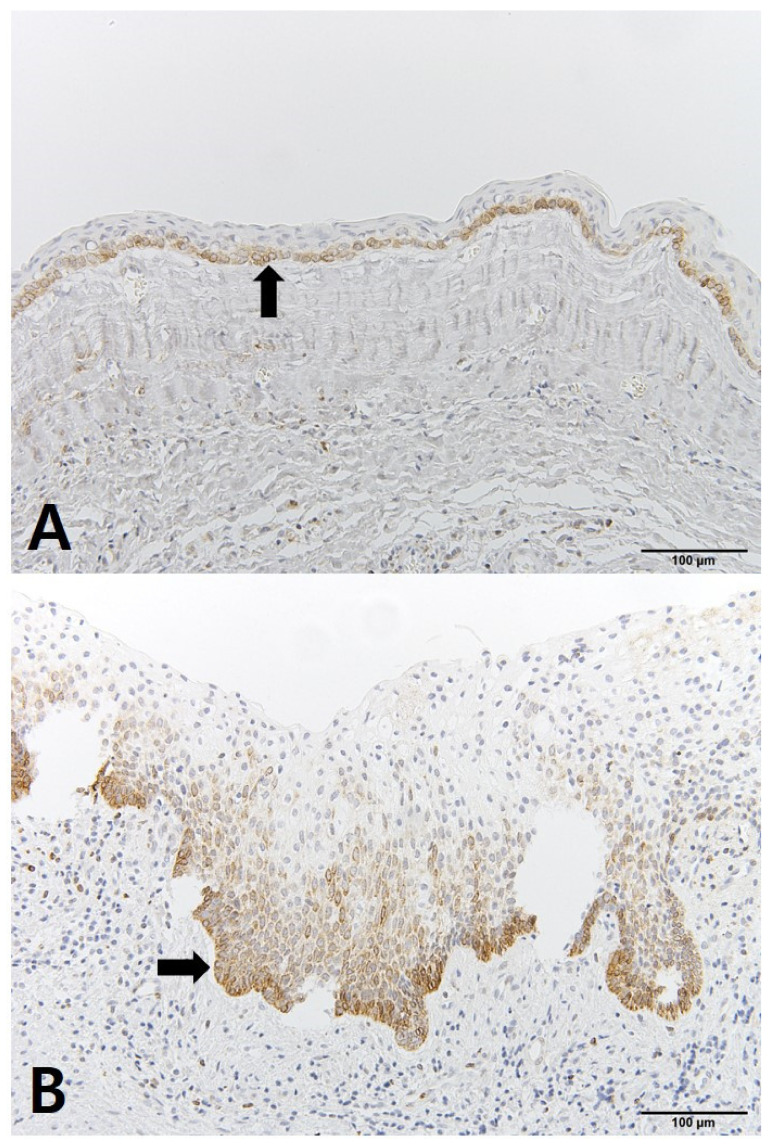
IHC staining of Bcl-2: (**A**) tissue obtained at decompression; (**B**) tissue obtained enucleation.

**Figure 5 jcm-10-00030-f005:**
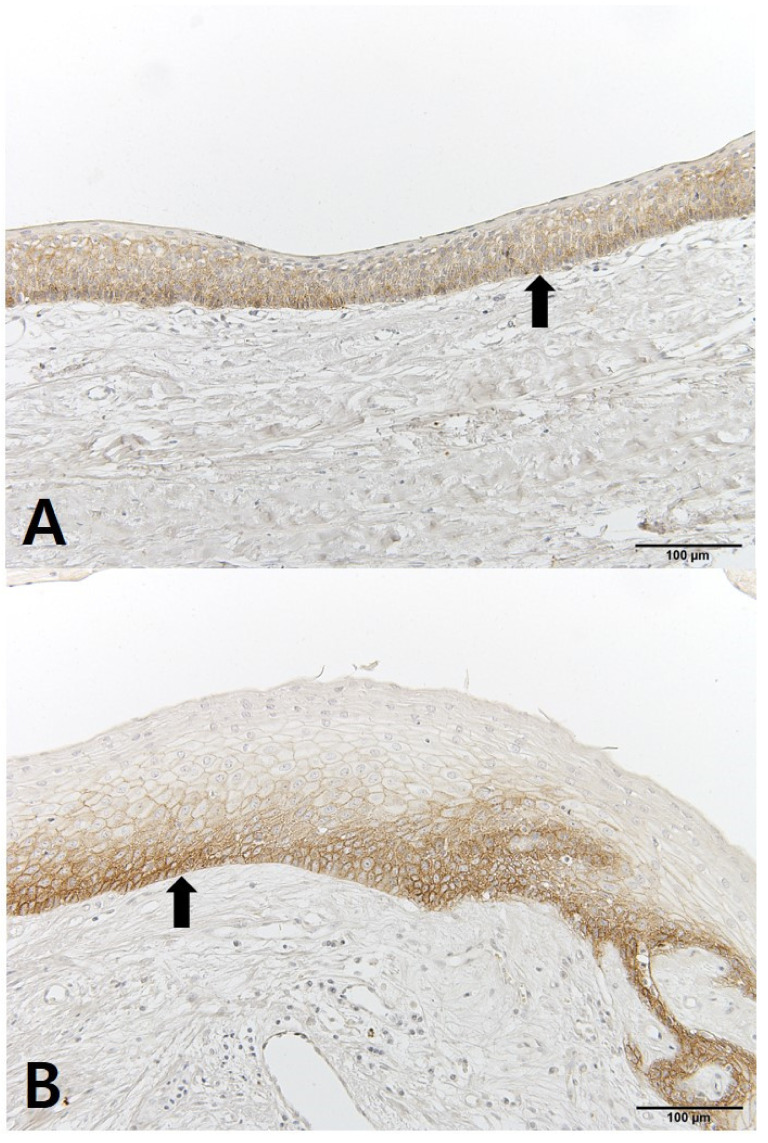
IHC staining of epidermal growth factor receptor (EGFR): (**A**) tissue obtained at decompression; (**B**) tissue obtained enucleation.

**Figure 6 jcm-10-00030-f006:**
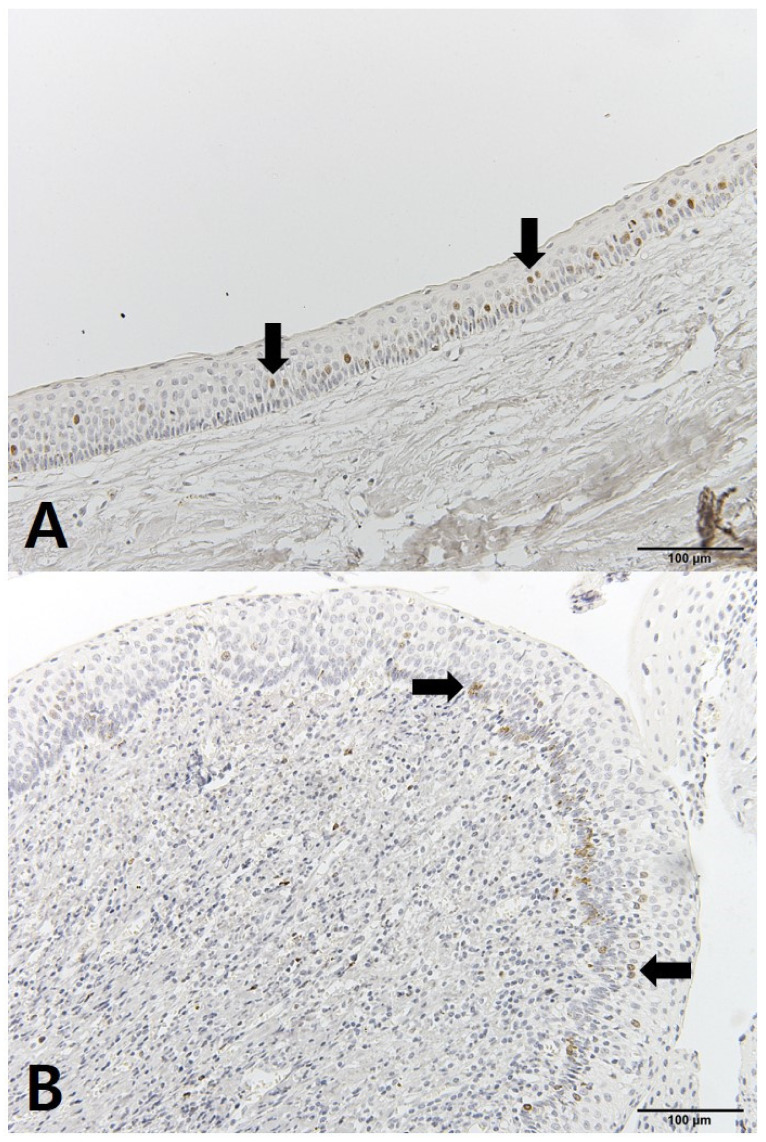
IHC staining of Ki-67: (**A**) tissue obtained at decompression; (**B**) tissue obtained enucleation.

**Figure 7 jcm-10-00030-f007:**
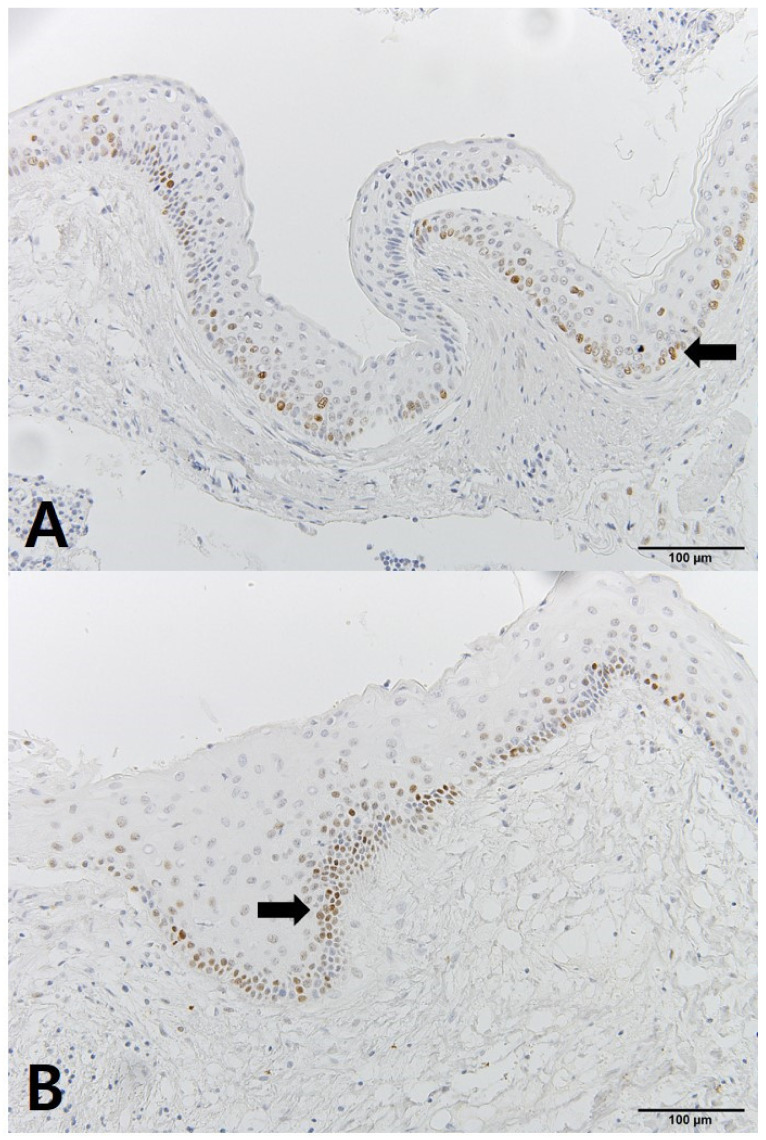
IHC staining of P53: (**A**) tissue obtained at decompression; (**B**) tissue obtained enucleation.

**Figure 8 jcm-10-00030-f008:**
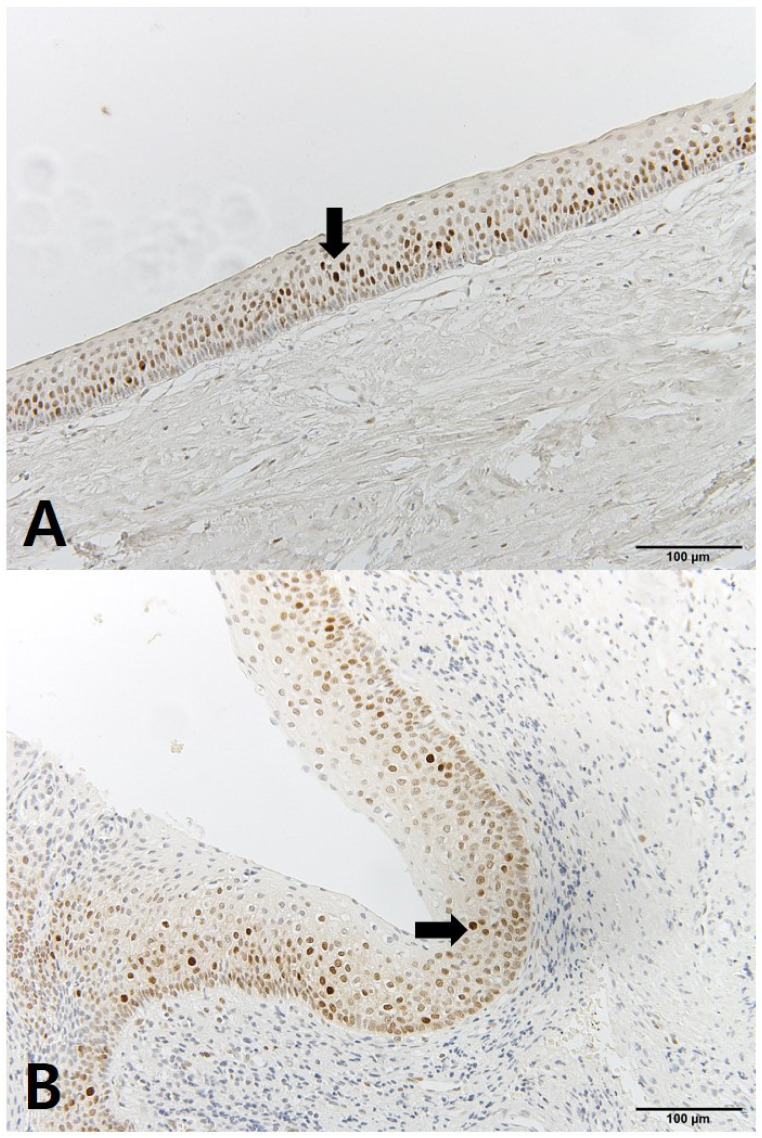
IHC staining of proliferating cell nuclear antigen (PCNA): (**A**) tissue obtained at decompression; (**B**) tissue obtained enucleation.

**Figure 9 jcm-10-00030-f009:**
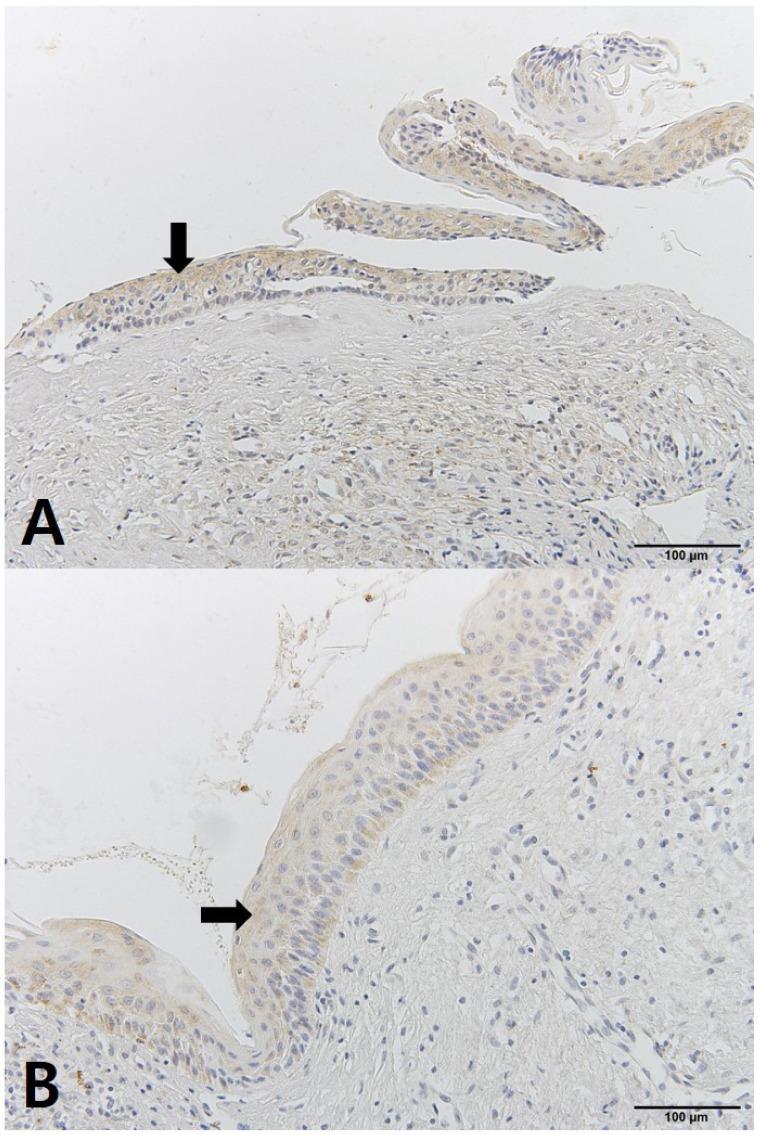
IHC staining of SMO: (**A**) tissue obtained at decompression; (**B**) tissue obtained enucleation.

**Table 1 jcm-10-00030-t001:** Patient information.

Patient Number	Age	Gender	Area	Adjacent Teeth	Decompression Period (Month)
1	38	Male	Maxilla Rt.	#18	5 m
2	32	Female	Maxilla Rt., Sinus	#18	12 m
3	36	Male	Maxilla Lt.		7 m
4	23	Female	Maxilla Lt., Sinus	#28	4 m 2 wk
5	24	Male	Maxilla Lt.	#28	5 m
6	23	Male	Maxilla Lt.	#28	10 m
7	20	Male	Maxilla Rt.	#18	11 m 2 wk
8	44	Female	Maxilla Lt., Sinus	#23, 24, 25	4 m
9	27	Female	Maxilla Lt.	#28	6 m
10	40	Male	Mandible Rt.		7 m 3 wk
11	59	Female	Mandible Rt.		8 m
12	22	Male	Mandible Lt.		8 m
13	68	Female	Mandible Rt., Ramus	#46, 47	6 m 3 wk
14	61	Male	Mandible Rt.	#38	9 m
15	61	Female	Mandible Rt.		8 m 2 wk
16	81	Female	Mandible Rt., Condyle, Ramus		5 m
17	19	Male	Mandible Lt.		7 m
18	32	Female	Mandible Rt., Ramus, body	#38	10 m
19	27	Male	Mandible Rt.,		4 m

m, month; wk, week.

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
