# Peer review of "Changes in Cellular Regulatory Factors before and after Decompression of Odontogenic Keratocysts"

_jcm, 2020, doi:10.3390/jcm10010030_

Round 1

Reviewer 1 Report

Inclusion/exclusion criteria have to be well depitcted and referred to patient

Add reference and discuss further 

Remove genetic information from IHC methods about the markers and discuss later in discussion paragraph

Please perform IHC expression according to Detre et al. A "quickscore" method for immunohistochemical semiquantitation: validation for oestrogen receptor in breast carcinomas

Improve statistical analysis: spearman/pearson correlation/ Mann-whitney, Kruskal-Wallis

Author Response

Reviewer #1: Inclusion/exclusion criteria have to be well depitcted and referred to patient. Add reference and discuss further

☞ We depicted inclusion/exclusion criteria in the manuscript, stating this in Materials & Method as The samples met the inclusion criteria of histologically-confirmed OKC which had been subjected to decompression treatment, followed by enucleation on the same lesion at later on. Cases of OKC associated with basal cell nevus syndrome, orthokeratinized OKC were excluded.”.

Reviewer #1: Remove genetic information from IHC methods about the markers and discuss later in discussion paragraph

☞ We agree with the reviewer’s opinion and removed the genetic information from IHC methods about the markers. It would be more appropriate to convey the role of each markers briefly in material and method section and explain about the detail information in the discussion section. In the revised manuscript, it is arranged in the discussion part as you suggested.

Reviewer #1: Please perform IHC expression according to Detre et al. A "quickscore" method for immunohistochemical semiquantitation: validation for oestrogen receptor in breast carcinomas

☞ The reason we used imageJ software program in our study was due to the absence of pathologist in our group. Many methods on quantifying IHC staining which include H-scoring system has a characteristic that professionalism and experiences are required. As we mentioned on discussion line 497 to line 503, it is confirmed throughout numerous studied to use software programs such as imageJ in numericalization of IHC. Fuhrich D. G. et al compared intraobserver and interobserver variation between traditional histological score (HSCORE) and digital HSCORE (D-HSCORE) performed by expert and naive researchers. Immunohistochemical analysis of β3 integrin subunit of 100 endometrial biopsies obtained from the midluteal phase of the menstrual cycle were reanalyzed using ImageJ software (D-HSCORE). The study concluded the D-HSCORE performed by an inexperienced researcher has high correlation to traditional HSCORE performed by an expert [71]. Similar results were yielded in other studies that compared manual counting with quantification using computer programs [72-74].

We are concerned that we will not be able to perform “quickscore” method to the study in our current state.

Reviewer #1: Improve statistical analysis: spearman/pearson correlation/ Mann-whitney, Kruskal-Wallis

☞ On this matter, I am currently having a discussion with a statistician. I apologize for taking time on statistic analysis, but as I arrive at a conclusion, I will send an answer again.

Reviewer 2 Report

The paper is very interesting. The authors did a complete literature review on the pathology discussed on the paper making a clear view of the pathology and treatment, that is not always described in the other works on this topic. The authors included also their personal case series, the number of patients it's not massive but it could be considered enough to have some realistic results from a clinical point of view and helped to have a stronger conclusion. The focused and give some good answer on the question of the topic of the paper

It could be interesting for the readers include some radiological image of the OKC in the introduction section to have a really complete idea of the pathology in every aspect also the clinical one.

Author Response

Reviewer #2: The paper is very interesting. The authors did a complete literature review on the pathology discussed on the paper making a clear view of the pathology and treatment, that is not always described in the other works on this topic. The authors included also their personal case series, the number of patients it's not massive but it could be considered enough to have some realistic results from a clinical point of view and helped to have a stronger conclusion. The focused and give some good answer on the question of the topic of the paper

It could be interesting for the readers include some radiological image of the OKC in the introduction section to have a really complete idea of the pathology in every aspect also the clinical one.

☞ We appreciate your reviewing our manuscript. As you have mentioned, radiological images of the OKC have been included to the revised manuscript as fig1.

Reviewer 3 Report

It is an interesting work, with an adequate and precise methodology, which deserves to be published.

The results, sometimes contradictory, previopusly indicated by other authors appear here well reviewed.

The article forces to resolve in the future with more extensive studies - with a well-planned methodology- the existing controversy on this issue.

Author Response

Reviewer #3: It is an interesting work, with an adequate and precise methodology, which deserves to be published.

The results, sometimes contradictory, previously indicated by other authors appear here well reviewed.

The article forces to resolve in the future with more extensive studies - with a well-planned methodology- the existing controversy on this issue.

☞ Thank you for helpful comments. As you mentioned, we hope that we will be able to embark on an expanded study on this subject.

Reviewer 4 Report

It is a well structured work but there are some aspects to be implemented.
On lines 320-325 you talk about "multiple studies" but
the bibliographic references of the various works that you are talking about are missing.
No other well-documented conservative systems are mentioned
to reduce the number of relapses of odontogenic keratocysts,
such as Carnoy's fluid and cryotherapy.
Please add these parts with bibliographic references.
It is important to stress more the concept of preferring decompression as a treatment,
compared to enucleation,
highlighting all the damage that could result documented in the literature:
it would give more value to the work you propose. The reference bibliography is in many cases rather dated,
please change the older bibliographic entries (1980-1996)
with more recent ones (last 5 years) where possible.
Kind Regards

Author Response

Reviewer #4: It is a well structured work but there are some aspects to be implemented.
On lines 320-325 you talk about "multiple studies" but the bibliographic references of the various works that you are talking about are missing.

☞ We appreciate your reviewing our manuscript. We mentioned the multiple studies on introduction part in line 67 to 69. Its aggressive clinical behavior and frequent recurrence following curettage has been the focus of several studies, which indicate that the OKC epithelial lining may have some intrinsic growth potential [8,10-14]. However, we missed to mark the bibliographic references on the later part. So we added the references to it.

Reviewer #4: No other well-documented conservative systems are mentioned to reduce the number of relapses of odontogenic keratocysts, such as Carnoy's fluid and cryotherapy. Please add these parts with bibliographic references.

☞ Thank you for helpful comments. We mentioned about various treatment modality on introduction part in line 71 to 84, and carnoy’s fluid as well. We included cryotherapy and reference [23]. Studies regarding to recurrence rate are mentioned on discussion part in line 475 to 483

Reviewer #4: It is important to stress more the concept of preferring decompression as a treatment, compared to enucleation, highlighting all the damage that could result documented in the literature: it would give more value to the work you propose.

☞ Thank you for helpful comments. In our department, in case of large lesions, decompression process is performed to reduce its size and later on mass excision or enucleation is done. But if the size of lesion is not so big, or if it is possible to perform mass excision or enucleation without giving damages to important structures such as inferior alveolar nerve, decompression is not always necessary. As we described in conclusion, there was no statistical evidence that decompression reduces the aggressive behavior of OKC within the epithelial cyst lining itself. This might indicate that decompression does not change the biological behavior of the epithelial cyst lining, as well as the recurrence rates.

Reviewer #4: The reference bibliography is in many cases rather dated, please change the older bibliographic entries (1980-1996) with more recent ones (last 5 years) where possible.

☞ As you suggested, we added more recent studies to the reference bibliography.